# WINNER-TAKE-ALL SPIKING TRANSFORMER FOR LANGUAGE MODELING

## ABSTRACT

Spiking Transformers, which combine the scalability of Transformers with the sparse, energy-efficient dynamics of Spiking Neural Networks (SNNs), have achieved strong results in neuromorphic and vision tasks and attracted increasing attention. However, existing directly trained spiking transformers primarily focus on vision tasks with encoder-only architectures. In language modeling, convergence relies heavily on softmax-based spiking self-attention, which incurs high energy costs and poses challenges for neuromorphic deployment. To address this issue, we introduce Winner-Take-All (WTA) mechanisms into spiking transformers and propose two novel softmax-free, spike-driven self-attention modules: WTA Spiking Self-Attention (WSSA) and Causal WTA Spiking Self-Attention (CWSSA). Based on these, we design WTA-based Encoder-only Spiking Transformer (WE-SpikingFormer) for masked language modeling and WTA-based Decoder-only Spiking Transformer (WD-SpikingFormer) for causal language modeling, systematically exploring direct-training-based softmax-free fully spike-driven transformers for natural language processing. Extensive experiments on 16 datasets spanning natural language understanding, question-answering tasks, and commonsense reasoning tasks validate the effectiveness of our approach and highlight the promise of spiking transformers for general language modeling and energy-efficient artificial intelligence.

## 1 INTRODUCTION

Spiking Neural Networks (SNNs), regarded as the third generation of neural networks (Maass, 1997), offer high biological plausibility and energy efficiency through their event-driven dynamics, making them strong contenders to Artificial Neural Networks (ANNs) (Roy et al., 2019). Specifically, by transmitting information through binary spikes, SNNs replace traditional high-power multiply-accumulate (MAC) operations with low-power accumulate (AC) operations, thereby achieving substantial energy savings.

Spiking transformers, which combine the architectural strengths of Transformers with the event-driven, sparse, and energy-efficient properties of spiking neural networks, have achieved great progress on both neuromorphic datasets and large-scale vision datasets (Zhou et al., 2023b;a; Yao et al., 2023a; 2024; Zhou et al., 2024; Yao et al., 2025) and have attracted significant attention. Spiking Transformers process information through discrete spikes with low-power operation, naturally capturing temporal dynamics that enhance biological plausibility and suitability for neuromorphic hardware. At the same time, they achieve performance comparable to that of ANN counterparts while offering superior energy efficiency.

Current directly trained spiking transformers have primarily targeted computer vision tasks (Zhou et al., 2023b;a; Yao et al., 2023a; 2024; Zhou et al., 2024; Yao et al., 2025) with encoder-only architectures. Spiking transformers with softmax-free self-attention function well in visual tasks because image patches are locally correlated and highly redundant. Repeated low-level features, such as edges and textures, allow softmax-free attention to achieve competitive performance.

By contrast, as the extension to spiking LLM is a growing trend, language signals are sparse and heavily depend on long-range dependencies, making softmax-free spiking transformers far more difficult to design for language modeling. SpikeBert (Lv et al., 2023) is an early attempt in this direction. As a softmax-free spiking transformer, SpikeBert adapts Spikformer (Zhou et al., 2023b)

for language tasks by proposing a two-stage knowledge distillation training method, and achieves 59.7% accuracy on the natural language understanding benchmark (GLUE dev datasets) (Wang et al., 2018). The substantial performance gap between SpikeBert and BERT (Devlin et al., 2019) (19.9%) highlights the difficulty of directly applying spiking self-attention from a vision-based spiking transformer (spikformer) to language modeling. More recently, SpikeLM (Xing et al., 2024b) proposes a spike formulation with bi-directional ternary firing for language modeling and achieves promising results in language modeling. SpikeLLM (Xing et al., 2024a) proposes generalized integrate-and-fire (GIF) neurons and an optimal brain spiking framework for spiking transformers. However, SpikeLM retains non-spiking activation GeLU (Hendrycks & Gimpel, 2016) in MLP blocks, and SpikeLLM retains SiLU (Hendrycks & Gimpel, 2016) in MLP blocks. In particular, both SpikeLM and SpikeLLM are softmax-based spiking transformers. The softmax with complex exponential and division operations brings huge challenges to energy consumption and neuromorphic deployment (Zhou et al., 2023b;a).

To address these limitations, we explore softmax-free fully spike-driven transformers, tailor-made for language modeling. Firstly, we introduce a brain-inspired mechanism: Winner-Take-All (WTA) for spiking transformers to replace the softmax operation in language modeling. Winner-take-all mimics biological lateral inhibition to enforce sparsity and focus attention on the most relevant tokens, serving as an extremely sparse alternative to softmax, which makes it particularly suitable for spiking language modeling. Through incorporating Winner-Take-All biological mechanisms, we developed two kinds of softmax-free, fully spike-driven transformers without floating-point multiplications: WTA-based Encoder-only SpikingFormer for masked language modeling and WTA-based Decoder-only SpikingFormer for causal language modeling. These architectures expand the spiking transformer family and advance both neuromorphic intelligence and energy-efficient artificial intelligence. Our main contributions are as follows :

1) Leveraging spike-driven and Winner-Take-All (WTA) biological mechanisms, we proposed two novel spike-driven self-attention for language modeling: WTA Spiking Self-Attention (**WSSA**) and Causal WTA Spiking Self-Attention (**CWSSA**). The self-attention floating-point multiplication operator between Query, Key, Value is replaced by sparse addition with high energy efficiency.

2) We developed a WTA-based Encoder-only Spiking Transformer (**WE-SpikingFormer**) with WSSA for masked language modeling, and a WTA-based Decoder-only Spiking Transformer (**WD-SpikingFormer**) with CWSSA for causal language modeling, systematically exploring direct-training-based softmax-free fully spike-driven transformers in language modeling.

3) We evaluate our models on 16 datasets covering natural language understanding, question-answering, and commonsense reasoning tasks. Extensive experiments demonstrate the promise of directly trained spiking transformers for general language modeling and energy-efficient artificial intelligence.

## 2 RELATED WORK

### 2.1 SPIKING TRANSFORMERS IN VISION TASKS

Spiking transformers with encoder-only architectures have been widely adopted, particularly in visual tasks (Zhou et al., 2023b;a; Yao et al., 2023a; 2024; Zhou et al., 2024; Yao et al., 2025). Spikformer (Zhou et al., 2023b) introduced Spiking Self-Attention (SSA), which replaces softmax with sparse spike-form Query, Key, and Value, achieving 74.81% accuracy on ImageNet-1k with only four time steps—the first strong evidence of the potential of transformer-based SNNs. Building on this, Spikingformer (Zhou et al., 2023a) employed a pre-activation shortcut to eliminate floating-point multiplications and reduce firing rates, further improving accuracy to 75.85%. Spike-driven Transformer (Yao et al., 2023a) proposed Spike-Driven Self-Attention (SDSA), which relies solely on masking and addition, reaching 77.07% on ImageNet-1k with significantly lower computational cost. Recently, hierarchical visual spiking transformers (Yao et al., 2024; Zhou et al., 2024; Yao et al., 2025) has achieved a performance of over 80% on ImageNet while maintaining high energy efficiency.

## 2.2 Spiking Transformers in Language tasks

SpikeBert (Lv et al., 2023) is a softmax-free spiking transformer, which improves Spikformer (Zhou et al., 2023b) to process language tasks and propose a two-stage knowledge distillation method for training it. The two-stage knowledge distillation method combines pre-training by distilling knowledge from BERT with a large collection of unlabelled texts and fine-tuning with task-specific instances via knowledge distillation from the BERT fine-tuned on the same training examples, and achieves 59.7% accuracy on the natural language understanding benchmark (GLUE dev datasets) (Wang et al., 2018), while also indicating that directly applying vision-based spiking self-attention leads to suboptimal performance on language tasks. SpikeGPT (Zhu et al., 2023) is an RWKV architecture-based spiking language model and retains the exponential and division operations similar to the softmax operation. SpikeLM (Xing et al., 2024b) demonstrates promising results in language modeling; however, they typically retain key components of vanilla attention, such as floating-point matrix multiplications, softmax operations, and non-spiking activation(GeLU (Hendrycks & Gimpel, 2016)) in MLP blocks. SpikeLLM (Xing et al., 2024a) is also a softmax-based decoder-only spiking transformer. Besides, SpikeLLM retains a large number of nonlinear operations of the llama (Touvron et al., 2023), such as SiLU (Hendrycks & Gimpel, 2016) non-spiking activation in the MLP block and retains rotary position embedding (Su et al., 2024) with non-spike-driven computation in Query and Key. By contrast, our proposed WE-SpikingFormer and WD-SpikingFormer eliminate these non-spiking components to realize a fully spike-driven transformer architecture. In short, this work focuses on designing softmax-free, energy-efficient spike-driven transformers for language modeling.

## 3 Method

### 3.1 Spiking Neuron Model

In this part, to explore more expressive yet efficient spiking neurons for language modeling, we adopt two variants of the Leaky Integrate-and-Fire (LIF) neuron model (Maass, 1997) for direct training: T-LIF (Xing et al., 2024b) and NI-LIF (Lei et al., 2025).

**Ternary-spiking-based Leaky Integrate-and-Fire (T-LIF) model.** Compared to the original LIF neuron, the ternary spiking in SpikeLM (Xing et al., 2024b) extends binary spikes $\{0, 1\}$ to ternary values $\{-\alpha, 0, \alpha\}$ based on membrane potential intensity. The dynamics of the Ternary spiking-based Leaky Integrate-and-Fire (T-LIF) model are formulated as:

$$U[t] = H[t-1] + X[t], \tag{1}$$

$$S[t] = \begin{cases} -1 \cdot \alpha[t], & \text{if } U[t] < -\alpha[t], \\ 0 \cdot \alpha[t], & \text{if } U[t] \in (-\alpha[t], +\alpha[t]), \\ +1 \cdot \alpha[t], & \text{if } U[t] > +\alpha[t], \end{cases} \tag{2}$$

$$H[t] = V_{\text{reset}} |b| + (\beta U[t])(1 - |b|), \tag{3}$$

where $X[t]$ is the input current at time step $t$, $S[t] = b \cdot \alpha[t]$ and $b \in \{-1, 0, 1\}$. $U[t]$ represents the membrane potential after the triggered event, which will decay directly to $H[t]$ if no spike is generated (where $\beta < 1$ is the decay factor) and otherwise equals to the reset potential $V_{reset}$.

**Normalized Integer Leaky Integrate-and-Fire (NI-LIF) Model.** This neuron adopts integer-training and spike-inference way (Luo et al., 2024), the dynamics of the normalized integer leaky integrate-and-fire model are formulated as follows:

$$U[t] = H[t-1] + X[t], \tag{4}$$

$$S[t] = \text{clip}(\text{round}(U[t]), 0, D)/D, \tag{5}$$

$$H[t] = \beta(U[t] - S[t] \times D), \tag{6}$$

where $\text{clip}(U[t], min, max)$ denotes the operation of clipping $U[t]$ to $[min, max]$, $D$ indicates the maximum quantized integer value and unfold into $D$ time steps when inference on neuromorphic chips. That is, the total time step T of the spike sequence is $T * D$, where $T$ is the normalized integer time step. For example, NI-LIF(1×4) unfolds into a binary spike sequence of time step T = 4. In our experiments, we apply T-LIF to WE-SpikingFormer and NI-LIF(1×4) to WD-SpikingFormer.

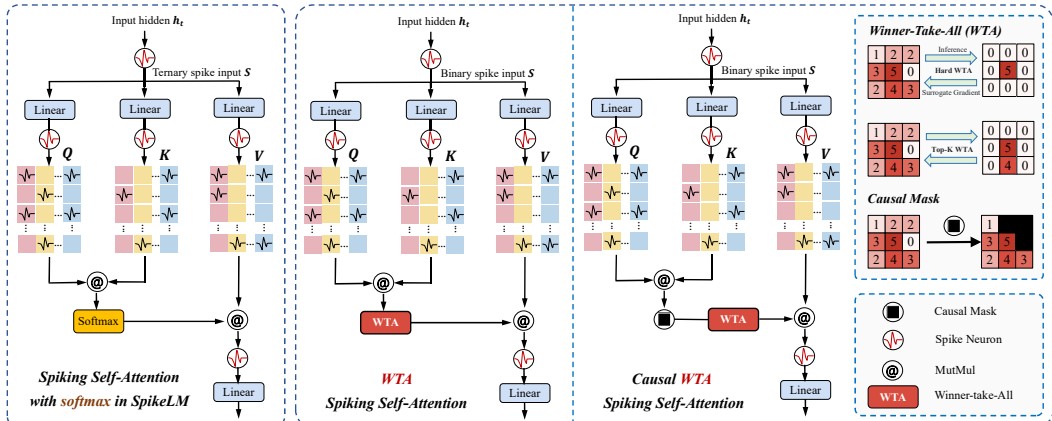

Figure 1: Overview of WTA-based Spiking Self-Attention (WSSA) and Causal WTA-based Spiking Self-Attention (CWSSA). The left shows the Softmax-based spiking self-attention in SpikeLM (Xing et al., 2024b). The right shows our WSSA in WE-SpikingFormer for masked language modeling and CWSSA in WD-SpikingFormer for causal language modeling.

## 3.2 WINNER-TAKE-ALL LAYER

In the visual system, multiple neurons are sensitive to different directions at the same location. The neurons with the strongest response (such as detecting vertical edges) will suppress other direction detectors, leaving only a dominant signal. The winner-take-all computation (Maass, 2000), inspired by this biological mechanism, is incorporated into spiking neural networks as a biologically plausible strategy for enforcing sparsity. It simulates the "lateral inhibition" phenomenon in biological nervous systems - that is, the strongest signal inhibits other competing signals, ultimately forming a single dominant response and ensuring that attention remains focused on the most relevant token. In this part, we mainly introduce the theory of WTA and its extensions.

**Hard WTA.** Hard WTA is an extremely sparse neural activation mechanism. In a group of competing units, only the unit with the highest activation value is allowed to retain the original output, and the outputs of all other units are forced to zero. Given an input vector $A = [a_1, \ldots, a_n]$, the output layer of WTA is $Y = [y_1, \ldots, y_n]$. Hard WTA can be formulated as follows:

$$y_i = \begin{cases} a_i & \text{if } i = \arg\max_{j \in (1, \ldots, N)} a_j, \\ 0 & \text{otherwise.} \end{cases} \tag{7}$$

**Top-K WTA.** As a sparse neural activation mechanism, Top-K WTA selects the K units with the highest scores from a set of input signals, retains only their original values, and sets the outputs of all other units to zero. Top-K WTA allows a limited number of "winners" to coexist, thus achieving a balance between sparsity and expressiveness. Hard WTA can be formulated as follows:

$$y_i = \begin{cases} a_i & \text{if } a_i \in \text{Top-k}(a_1, \ldots, a_n), \\ 0 & \text{otherwise.} \end{cases} \tag{8}$$

**Sparsemax.** Sparsemax is a differentiable, adaptively sparse neural mechanism that can automatically generate some zero values in the output; that is, only a few significant elements are given non-zero probability, and the remaining elements are precisely truncated to zero, thereby achieving adaptive sparsity. Sparsemax can be formulated as follows:

$$y_i = \max\{a_i - \tau, 0\}, \quad i = 1, \ldots, N. \tag{9}$$

Eq. 9 needs to satisfy $\sum_i y_i = 1$, and $\tau$ is the adaptive threshold.

**Surrogate Gradient for WTA Layer.** The Winner-Take-All mechanism is non-differentiable because it produces discrete, discontinuous outputs with zero gradients almost everywhere and undefined gradients at decision boundaries. Therefore, we choose the surrogate gradient for winner-take-all

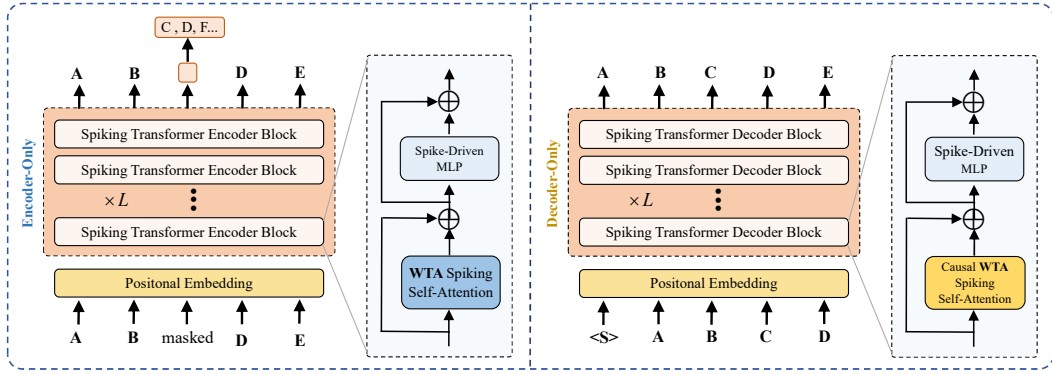

Figure 2: The overview of WE-SpikingFormer and WD-SpikingFormer. The left shows WE-SpikingFormer (WTA-based Encoder-only Spiking Transformer) for spike-based masked language modeling. The right shows WD-SpikingFormer (WTA-based Decoder-only Spiking Transformer) for spike-based causal language modeling.

during training, which does not affect the high energy efficiency of the model during inference. Specifically, we use the gradient of softmax to approximate the "gradient" of the winner-take-all layer, which is shown as follows:

$$\frac{\partial y}{\partial a} = \text{softmax}'(a). \tag{10}$$

where $\text{softmax}'(.)$ is the derivative of the softmax function.

Winner-take-all is functionally equivalent to an extremely sparse version of softmax, ensuring that attention remains focused on the most relevant token. It uses neural dynamics lateral inhibition instead of exponential normalization. It is more suitable for spiking language modeling due to its sparse dependencies and lateral inhibition, while also offering energy efficiency advantages. In our experiments, we exploit Hard WTA by default owing to its superior energy efficiency and accuracy.

### 3.3 CAUSAL WTA SPIKING SELF-ATTENTION

In this part, we propose two self-attention mechanisms based on the winner-take-all layer for Spiking language modeling: WTA Spiking Self-Attention (WSSA) and Causal WTA Spiking Self-Attention (CWSSA), which are shown in Fig. 1.

**Causal WTA Spiking Self-Attention (CWSSA)** is a spike-driven self-attention for spike-based causal language modeling with Decoder-only Transformers. CWSSA can be formulated as follows:

$$\mathbf{X}' = \text{SN}(\mathbf{X}), \tag{11}$$

$$\mathbf{Q} = \text{SN}_Q(\text{Linear}_Q(\mathbf{X}')), \tag{12}$$

$$\mathbf{K} = \text{SN}_K(\text{Linear}_K(\mathbf{X}')), \tag{13}$$

$$\mathbf{V} = \text{SN}_V(\text{Linear}_V(\mathbf{X}')), \tag{14}$$

$$A_\text{w}(\mathbf{Q}, \mathbf{K}) = \text{WTA}(\text{CausalMask}(\mathbf{Q}\mathbf{K}^\text{T} * s)), \tag{15}$$

$$\text{Out}(A_\text{w}, \mathbf{V}) = \text{Linear}(\text{SN}(A_\text{w}\mathbf{V})), \tag{16}$$

where $s$ is the scaling factor, same in (Zhou et al., 2023b), $A_\text{w}$ is the attention weights, and SN means the spiking neuron.

**WTA Spiking Self-Attention (WSSA)** is a spike-driven self-attention for spike-based masked language modeling with Encoder-only Transformers. The attention mechanism is similar to CWSSA, with the main difference that the causal mask in Eq. 16 is removed:

$$A_\text{w}(\mathbf{Q}, \mathbf{K}) = \text{WTA}((\mathbf{Q}\mathbf{K}^\text{T} * s)). \tag{17}$$

## 3.4 Language Modeling

Based on WSSA and CWSSA in Section 3.3, we built two spiking transformers for spike-based language modeling: WTA-based Spiking Encoder (WE-SpikingFormer) and WTA-based Spiking Decoder (WD-SpikingFormer) in this part.

**Model Architecture.** The overview of WE-SpikingFormer and WD-SpikingFormer is shown in Fig. 2. The decoder block in WD-SpikingFormer can be formulated as follows:

$$X'_l = \text{CWSSA}\left(X_{l-1}\right) + X_{l-1}, \quad X'_l \in \mathbb{R}^{T \times d}, l = 1 \ldots L, \tag{18}$$

$$X_l = \text{SMLP}\left(X'_l\right) + X'_l, \quad X_l \in \mathbb{R}^{T \times d}, l = 1 \ldots L, \tag{19}$$

where SMLP means Spike-driven Multi-Layer Perceptron, which is realized by {SN-Linear-SN-Linear} sequence. The encoder block in WE-SpikingFormer is similar to the decoder one in WD-SpikingFormer, with the main difference that CWSSA in Eq. 18 is replaced by WSSA in Eq. 20:

$$X'_l = \text{WSSA}\left(X_{l-1}\right) + X_{l-1}, \quad X'_l \in \mathbb{R}^{T \times d}, l = 1 \ldots L. \tag{20}$$

**Masked Language Modeling.** Given a sequence of tokens $\text{x} = (x_1, x_2, \ldots, x_T)$, randomly select a portion of the locations $M \subseteq \{1, 2, \ldots, T\}$ for masking. The masked sequence is obtained: $\tilde{x} = (\tilde{x}_1, \tilde{x}_2, \ldots, \tilde{x}_T)$. The goal of Masked Language Modeling (MLM) (Devlin et al., 2019) is to let the model predict the masked tokens based on the unmasked tokens, which can be formulated as follows:

$$p\left(x_M \mid \tilde{x}_{\setminus M}\right), \tag{21}$$

where $x_M = \{x_t : t \in M\}$ denotes the masked token, $\tilde{x}_{\setminus M}$ denotes the visible tokens that are retained. Thus, the loss function of MLM can be formulated as follows:

$$\mathcal{L}_{\text{MLM}} = -\sum_{t \in M} \log p_\theta\left(x_t \mid \tilde{x}\right). \tag{22}$$

In our work, we apply WE-SpikingFormer for masked language modeling for pretraining.

**Causal Language Modeling.** Causal Language Modeling (CLM) (Radford et al., 2018; Ouyang et al., 2022) is the most widely used language modeling method. Its core idea is autoregressive generation: when predicting the next word, only the context before the current position is allowed to be used, and future words cannot be accessed. Given a sequence of tokens $\text{x} = (x_1, x_2, \ldots, x_T)$. The training strategy of CLM can be formulated as follows:

$$p(x) = \prod_{t=1}^{T} p\left(x_t \mid x_{<t}\right), \tag{23}$$

where $x_t$ represents the $t$-th token, $x_{<t} = (x_1, x_2, \ldots, x_{t-1})$ represents all previous tokens. The optimization goal of CLM is to maximize its conditional probability decomposition. In network modeling, the causal mask is used to ensure the model only focuses on the previous tokens when calculating attention. The loss function of CLM can be formulated as follows:

$$\mathcal{L}_{\text{CLM}} = -\sum_{t=1}^{T} \log p_\theta\left(x_t \mid x_{<t}\right). \tag{24}$$

In our work, we apply WD-SpikingFormer for causal language modeling for pretraining.

## 4 Experiments

### 4.1 Natural Language Understanding

We evaluate WE-SpikingFormer on the standard GLUE (General Language Understanding Evaluation) benchmark (Wang et al., 2018), which is a widely adopted collection of datasets designed to evaluate and advance natural language understanding capabilities of machine learning models. GLUE contains 8 subsets for classification and regression, including single-sentence classification (CoLA,

Table 1: The results on the Natural Language Understanding task (GLUE datasets). "Avg." denotes "Average Accuracy (%)". The results of LIF-BERT, PSN-BERT, and SpikeLM are reported in Xing et al. (2024b).

| Model | T | MNLI | QQP | QNLI | SST-2 | CoLA | STS-B | MRPC | RTE | Avg. |
|---|---|---|---|---|---|---|---|---|---|---|
| BERT$_{base}$ | – | 83.4 | 71.2 | 90.5 | 93.5 | 52.1 | 85.8 | 88.9 | 66.4 | 79.6 |
| Q2BERT | – | 47.3 | 67.0 | 61.3 | 80.6 | 0.0 | 4.7 | 81.2 | 52.7 | 49.1 |
| BERT$_{3L}$ | – | 77.1 | 85.2 | 85.8 | 88.1 | 31.7 | 85.7 | 86.4 | 66.4 | 75.9 |
| SpikeLM | 4 | 77.2 | 83.9 | 85.3 | 87.0 | 38.8 | 84.9 | 85.7 | 69.0 | 76.5 |
| LIF-BERT | 4 | 35.2 | 0 | 50.5 | 50.9 | 0 | 0 | 81.2 | 52.7 | 34.6 |
| PSN-BERT | 4 | 35.2 | 0 | 50.5 | 50.9 | 0 | 6.8 | 81.2 | 52.7 | 34.7 |
| SpikeBERT | 4 | 71.0 | 68.2 | 66.4 | 85.4 | 16.9 | 18.7 | 82.0 | 57.5 | 59.7 |
| **WE-SpikingFormer** | 4 | 70.1 | 85.1 | 77.5 | 89.0 | 27.9 | 42.8 | 81.6 | 55.9 | **66.3** |

SST-2), pairwise sentence comparison (MPRC, QQP, RTE), sentence similarity (STS-B), and natural language inference (MNLI, QNLI).

We pretrain WE-SpikingFormer on Wikipedia-English (Devlin et al., 2019) with masked language modeling (Devlin et al., 2019) using 8 GPUs, and subsequently fine-tune it on the GLUE dev set. The ANN baseline includes BERT (Devlin et al., 2019) and Q2BERT (Zhang et al., 2020), the latter of which employs 2-bit weights and 8-bit activations.

The experimental results are shown in Table 1 and all models in Table 1 except BERT$_{3L}$ have 12 encoder blocks and have the same model size with about 0.1B parameters. WE-SpikingFormer achieves 66.3% average accuracy. In comparison, SpikeLM is a softmax-based spiking transformer and retains non-spiking activation GeLU (Hendrycks & Gimpel, 2016) in MLP blocks. These factors led to its relatively higher performance. As a softmax-free, fully spike-driven transformer, WE-SpikingFormer outperforms other spike-driven methods (LIF-BERT, PSN-BERT, and SpikeBERT) by a significant margin. WE-SpikingFormer vs. SpikeBERT vs. LIF-BERT. Acc: 66.3% vs. 59.7% vs. 34.6%.

## 4.2 QUESTION ANSWERING TASKS

In this experiment, we use the FineWeb-Edu (Lozhkov et al., 2024) dataset, a high-quality subset of the FineWeb corpus curated for factual and educational content, and sample 1B tokens from it for pretraining WD-SpikingFormer across 8 GPUs. We evaluate the model performance of WD-SpikingFormer on a diverse set of Question-Answering Tasks (QAT), including ARC-e (Clark et al., 2018), ARC-c (Clark et al., 2018), BoolQ (Clark et al., 2019), HeadQA (Vilares & Gómez-Rodríguez, 2019), and OpenBookQA (OBQA) (Mihaylov et al., 2018). These tasks measure the generalization and reasoning abilities without task-specific finetuning.

Table 2: The results on Question Answering Tasks (QAT). "Avg." denotes "Average Accuracy (%)". "E (mJ)" means "Energy consumption (mJ)". "T" means time step.

| Model | E (mJ) | T | ARC-e | ARC-c | BoolQ | HeadQA | OBQA | Avg. |
|---|---|---|---|---|---|---|---|---|
| SpikeLLM-7B | - | 4 | 31.3 | 23.6 | 53.8 | - | - | - |
| DeepSeek-Distill-Qwen-1.5B | 3398.3 | - | 26.2 | 26.9 | 60.3 | 25.7 | 27.1 | 33.2 |
| **WD-SpikingFormer-0.4B** | 238.4 | 4 | 30.0 | 22.3 | 37.8 | 26.0 | 26.1 | **28.4** |

The experimental results are shown in Table 2. On one hand, since SpikeLLM (Xing et al., 2024a) does not open-source the model file nor report energy consumption and the results of HeadQA and OBQA, our comparative analysis is limited to the average accuracy on the ARC-e, ARC-c, and HeadQA benchmarks. Despite being 17.5 times smaller (0.4B vs. 7B parameters), WD-SpikingFormer-0.4B achieves a competitive accuracy of 30.0%, approaching the 36.2% accuracy of the much larger

SpikeLLM-7B. On the other hand, WD-SpikingFormer-0.4B reduces energy consumption by an order of magnitude (238.4 mJ vs. 3398.3 mJ, only $7\%$ of the energy) while maintaining a small accuracy gap ($28.4\%$ vs. $33.2\%$) against DeepSeek-Distill-Qwen-1.5B, showcasing a superior energy-accuracy trade-off. The two comparisons show that our model WD-SpikingFormer-0.4B enjoys high energy efficiency.

### 4.3 COMMONSENSE REASONING TASKS

In this experiment, we evaluated the model performance of WD-SpikingFormer on a diverse set of Commonsense Reasoning Tasks (CRT), including the HellaSwag (Zellers et al., 2019), PIQA (Bisk et al., 2020), and Winograd (Sakaguchi et al., 2021) datasets. The pretraining process is the same as shown in Section 4.2.

Table 3: The results on Commonsense Reasoning Tasks. "Avg." denotes "Average Accuracy ($\%$)". "E (mJ)" means "Energy consumption (mJ)". "T" means time step.

| Model | E (mJ) | T | HellaSwag | PIQA | Winograd | Avg. |
|---|---|---|---|---|---|---|
| SpikeLLM-7B | - | 4 | 33.9 | 53.4 | 51.5 | 46.3 |
| DeepSeek-Distill-Qwen-1.5B | 3398.3 | - | 26.5 | 53.7 | 52.8 | 44.3 |
| **WD-SpikingFormer-0.4B** | 238.4 | 4 | 25.9 | 53.4 | 50.2 | **43.2** |

The experimental results are shown in Table 3. WD-SpikingFormer-0.4B uses smaller parameters (0.4B vs. 1.5B and 7B) while its accuracy remains close to the larger models ($43.2\%$ vs. $44.3\%$ and $46.3\%$). These results highlight that WD-SpikingFormer-0.4B delivers competitive performance despite its much smaller size.

### 4.4 ABLATION STUDY

We carry out the ablation study on the natural language understanding task (GLUE datasets), Question-Answering Tasks (QAT), and Commonsense Reasoning Tasks (CRT). The experimental results are shown in Table 4 and Table 5.

The ablation study on GLUE datasets is shown in Table 4. The three attention versions use the same model architecture, training process, and other experimental settings. We can find that the model with vision-oriented spiking self-attention ($Q\,K^T\,V*s$) achieves only $55.4\%$ accuracy on the GLUE set. However, the model in the softmax-based version ($\mathrm{Softmax}(Q\,K^T*s)\,V$) achieves only $66.8\%$ accuracy, which shows that the softmax layer has a great influence on language modeling, and vision-oriented spiking self-attention is not suitable for language modeling. The model (WE-SpikingFormer) in the Winner-Take-All version ($\mathrm{HardWTA}(Q\,K^T*s)\,V$), achieves $66.3\%$ accuracy, which is very close to the performance of the model in the softmax-based version. These experiments fully verify the effectiveness of our WTA-based spiking self-attention in masked language modeling.

Table 4: Ablation study of Winner-Take-All layer on natural language understanding tasks (GLUE dev set). "Avg." denotes "Average Accuracy ($\%$)". We apply the vision-oriented spiking self-attention (Zhou et al., 2023b) ($Q\,K^T\,V*s$) and softmax-based version ($\mathrm{Softmax}(Q\,K^T*s)\,V$) (Xing et al., 2024b) in this language task.

| Model | T | MNLI | QQP | QNLI | SST-2 | CoLA | STS-B | MRPC | RTE | Avg. |
|---|---|---|---|---|---|---|---|---|---|---|
| $Q\,K^T\,V*s$ | 4 | 67.6 | 70.1 | 68.9 | 80.3 | 8.9 | 16.7 | 79.4 | 51.3 | 55.4 |
| $\mathrm{Softmax}(Q\,K^T*s)\,V$ | 4 | 72.5 | 84.7 | 76.0 | 87.2 | 24.4 | 54.5 | 79.7 | 55.6 | 66.8 |
| $\mathrm{HardWTA}(Q\,K^T*s)\,V$ | 4 | 70.1 | 85.1 | 77.5 | 89.0 | 27.9 | 42.8 | 81.6 | 55.9 | **66.3** |

Table 5: Ablation study of Winner-Take-All layer and time steps on Question-Answering Tasks (QAT), and Commonsense Reasoning Tasks (CRT). Note that "✗" means the model does not converge in pretraining. "T" means time step. "Avg." denotes "Average Accuracy".

| Spiking Transformer | T | Method | QAT Avg.(%) | CRT Avg.(%) |
|---|---|---|---|---|
| | 4 | $Q\,K^T\,V*s$ | ✗ | - |
| | 4 | $\text{Softmax}(Q\,K^T*s)\,V$ | 28.7 | - |
| | 4 | $\text{HardWTA}(Q\,K^T*s)\,V$ | 28.4 | - |
| Decoder-Only | 4 | $Q\,K^T\,V*s$ | - | ✗ |
| | 4 | $\text{Softmax}(Q\,K^T*s)\,V$ | - | 43.3 |
| | 4 | $\text{HardWTA}(Q\,K^T*s)\,V$ | - | 43.2 |
| | 4 | $\text{Top-kWTA}(Q\,K^T*s)\,V$ | - | 42.9 |
| | 4 | $\text{SparseMax}(Q\,K^T*s)\,V$ | - | 43.2 |
| | 8 | $\text{HardWTA}(Q\,K^T*s)\,V$ | - | 43.7 |
| | 2 | $\text{HardWTA}(Q\,K^T*s)\,V$ | - | 40.1 |

The ablation study on question-answering tasks and commonsense reasoning tasks is shown in Table 5. The base experimental backbone is the Decoder-only spiking transformer. By combining the results in Table 4, the experimental results on the three tasks show that: 1) Vision-oriented spiking self-attention $(Q\,K^T\,V*s)$ does not work well in language modeling, often leading to convergence difficulties or suboptimal performance. $\text{HardWTA}(Q\,K^T*s)\,V$ vs. $(Q\,K^T\,V*s)$. GLUE: 66.3% vs. 55.4%; QAT: 28.4% vs. Not converging in pretraining; CRT: 43.2% vs. Not converging in pretraining. 2) The winner-take-all layer can approximately replace the softmax layer in self-attention computation of spiking transformers in language modeling while enjoying high energy efficiency during inference. $\text{HardWTA}(Q\,K^T*s)\,V$ vs. $\text{Softmax}(Q\,K^T*s)\,V$. GLUE: 66.3% vs. 66.8%; QAT: 28.4% vs. 28.7%; CRT: 43.2% Vs. 43.3%. The three tasks ablation study verified the effectiveness of WTA's lateral inhibition, tailor-made for the spiking transformers in language modeling. Furthermore, the performance of Hard WTA, Top-k WTA, and Sparsemax is very similar on CRT (43.2% vs. 42.9% vs. 43.2%), making Hard WTA the more cost-effective option. When the time step is increased to 8, the performance of the WTA-based decoder-only spiking transformer (WD-SpikingFormer-0.4B) further improves to an average accuracy of 43.7%.

## 5  CONCLUSION

In this work, we explore softmax-free fully spike-driven transformers for language modeling by introducing Winner-Take-All (WTA) mechanisms into spike-driven self-attention. We proposed two novel attention modules, WTA-based Spiking Self-Attention (WSSA), Causal WTA-based Spiking Self-Attention(CWSSA), and designed WE-SpikingFormer for masked language modeling and WD-SpikingFormer for causal language modeling. Our approach systematically extends directly trained spiking transformers from vision to language. Extensive experiments on 16 datasets spanning natural language understanding, question-answering tasks, and commonsense reasoning tasks validate the effectiveness of our models and highlight the potential of spiking transformers as a foundation for biologically inspired, energy-efficient, and general-purpose language modeling.

**Limitation.** This work first explores the direct-training-based, softmax-free, fully spike-driven transformers in language modeling, including masked language modeling and causal language modeling. A limitation of our study is the lack of exploration across different model scales, which we leave for future work.

## ETHICS STATEMENT

All experiments in this work are conducted on publicly available datasets without involving private or sensitive information. The proposed methods are intended purely for academic research, and any deployment should carefully consider potential ethical risks such as bias or misuse.

## REPRODUCIBILITY STATEMENT

The experimental results in this paper are reproducible. We describe the model architecture and training process details in the main text and appendix. We will release the source code after review.

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

# A   APPENDIX

## A.1   THE USE OF LARGE LANGUAGE MODELS (LLMS)

In this work, we used Large Language Models (LLMs) in a limited and auxiliary capacity. Specifically, the use of large models is mainly used to improve writing in this work. LLMs were not involved in designing algorithms, implementing models, or analyzing experimental results. All methodological innovations are independently conceived, implemented, and validated by the authors. Thus, the role of LLMs was restricted to improving manuscript writing, without influencing the scientific contributions of this paper.

## A.2   DATASET INTRODUCTION

**General Language Understanding Evaluation (GLUE).** GLUE benchmark (Wang et al., 2018) is a widely adopted framework for assessing natural language understanding (NLU) models across a variety of tasks, including single-sentence classification, sentence-pair classification, and linguistic acceptability. In our experiments, we focus on eight representative datasets within GLUE: MNLI (Multi-Genre Natural Language Inference): Predict whether a hypothesis is entailed, contradicted, or neutral with respect to a given premise across multiple text genres. QQP (Quora Question Pairs): Detect whether two questions from Quora convey the same meaning. QNLI (Question Natural Language Inference): Reformulated from QA, determines whether a sentence contains the answer to a question. SST-2 (Stanford Sentiment Treebank): Perform binary sentiment classification on movie reviews. CoLA (Corpus of Linguistic Acceptability): Judge whether a sentence is grammatically acceptable. STS-B (Semantic Textual Similarity Benchmark): Measure sentence-level semantic similarity on a scale from 0 to 5. MRPC (Microsoft Research Paraphrase Corpus): Identify whether two sentences are paraphrases. RTE (Recognizing Textual Entailment): Decide whether a hypothesis can be inferred from a premise, based on multiple entailment datasets.

**Question Answering Tasks (QAT).** ARC-e (Clark et al., 2018) and ARC-c (AI2 Reasoning Challenge, easy and challenge subsets) assess scientific knowledge and reasoning skills, with ARC-e focusing on simpler multiple-choice science questions and ARC-c including more complex ones that require advanced reasoning. BoolQ (Boolean Questions) (Clark et al., 2019) is a yes/no question-answering dataset derived from natural queries, requiring models to determine the truthfulness of a statement given a supporting passage. HeadQA (Head Question Answering) (Vilares & Gómez-Rodríguez, 2019) is a multilingual medical question-answering benchmark composed of exams for healthcare professionals, testing domain-specific knowledge. OBQA (OpenBookQA) (Mihaylov et al., 2018) evaluates a model's ability to answer elementary science questions by combining provided core facts with external common knowledge.

**Commonsense Reasoning Tasks (CRT).** HellaSwag (Zellers et al., 2019) is a large-scale dataset for grounded commonsense inference, where models must select the most plausible continuation

of a given context. PIQA (Physical Interaction Question Answering) (Bisk et al., 2020) focuses on physical commonsense reasoning, requiring models to choose the more feasible solution to everyday tasks. Winograd (Winograd Schema Challenge) (Sakaguchi et al., 2021) is a coreference resolution benchmark designed to test commonsense reasoning by requiring models to resolve pronoun references that cannot be disambiguated by syntax alone.

## A.3 ENERGY CONSUMPTION

SNNs replace traditional multiply-accumulate (MAC) operations with low-power accumulate (AC) operations. For ANNs, the overall energy consumption can be directly evaluated by their MACs. For example, given a linear layer with input dimension $m$ and output dimension $n$, its energy consumption can be calculated by:

$$E_{\text{Linear}}^{\text{Ann}} = m \times n \times E_{MAC}, \tag{25}$$

SNNs convert MAC-based matrix multiplications to pure accumulate operations (ACs). For SNNs, given the same example as the ANN case, the theoretical energy consumption of the linear layer can be calculated by:

$$E_{\text{Linear}}^{\text{Snn}} = m \times n \times E_{AC} \times fr \times T, \tag{26}$$

where $fr$ is the firing rate of the layer, and $T$ is the simulation time step of the spiking neuron. Refer to previous works(Kundu et al., 2021; Zhou et al., 2023b;a; Panda et al., 2020; Yao et al., 2023b). We assume that the MAC and AC operations are implemented on the 45nm hardware (Horowitz, 2014), where $E_{MAC} = 4.6pJ$ and $E_{AC} = 0.9pJ$. The energy consumption of the models in the paper is calculated by reasoning about 512 tokens.

## A.4 VISION-ORIENTED SPIKING SELF-ATTENTION

Vision-oriented Spiking Self-Attention (Zhou et al., 2023b;a) use a sparse spike-form $\mathbf{Q}, \mathbf{K}, \mathbf{V}$ for vision task modeling without softmax operation and floating-point matrix multiplication. The calculation process of vision-oriented spiking self-attention is formulated as follows:

$$\mathbf{I} = \text{SN}_I \left( \text{BN}_I \left( \mathbf{X}(\mathbf{W}_I) \right) \right), \ \mathbf{I} \in (\mathbf{Q}, \mathbf{K}, \mathbf{V}), \tag{27}$$

$$\text{SSA}'(\mathbf{Q}, \mathbf{K}, \mathbf{V}) = \text{SN} \left( \mathbf{Q} \mathbf{K}^{\text{T}} \mathbf{V} * s \right), \tag{28}$$

where $\mathbf{Q}, \mathbf{K}, \mathbf{V} \in \mathcal{R}^{T \times N \times D}$, the spike-form $\mathbf{Q}, \mathbf{K}, \mathbf{V}$ are computed by learnable linear layers. $s$ is a scaling factor. SN means spiking neuron layer. The calculation of SSA avoids floating-point multiplication, meeting the property of SNNs.

