# OpenReview forum: "Winner-Take-All Spiking Transformer for Language Modeling"
_ICLR.cc/2026/Conference — ICLR 2026 Conference Withdrawn Submission_

### Official Review · Reviewer_svxs · 2025-10-31

**Soundness:** 2
**Presentation:** 2
**Contribution:** 2
**Rating:** 4
**Confidence:** 3

**Summary:**

This paper introduces a Winner-Take-All mechanism to replace softmax in spiking transformers, aiming to make language models both energy-efficient and biologically plausible. The authors design two fully spike-driven architectures, WE-SpikingFormer for masked LM and WD-SpikingFormer for causal LM, and show that these models achieve competitive results on GLUE, QA, and commonsense reasoning tasks while greatly reducing energy consumption.

**Strengths:**

1. Replacing softmax with a biologically inspired WTA mechanism in spiking transformers is creative and technically interesting.
2. The authors evaluate across 16 diverse NLP datasets and include ablation studies to isolate the effect of WTA.

**Weaknesses:**

1. Despite the energy gains, accuracy still lags behind conventional transformers. The practical competitiveness remains unclear.

2. Only small-to-medium models (0.4B parameters) are tested; scaling trends are not explored.

3. WTA introduces non-differentiable operations, and while surrogate gradients are mentioned, there’s little analysis of convergence behavior or gradient quality.

4. The paper lacks a deeper analysis of why WTA maintains or improves attention quality beyond empirical results.

**Questions:**

Can WTA-based models scale to multi-billion parameter sizes without losing stability or accuracy?

---

### Official Review · Reviewer_Hdmf · 2025-11-01

**Soundness:** 3
**Presentation:** 2
**Contribution:** 2
**Rating:** 4
**Confidence:** 4

**Summary:**

This paper explores softmax-free, fully spike-driven Transformers for language modeling by integrating Winner-Take-All (WTA) mechanisms inspired by biological lateral inhibition. The authors propose WTA-based spiking self-attention modules and design both encoder-only and decoder-only spiking Transformers tailored for masked and causal language modeling. Their approach eliminates costly floating-point operations, achieving efficient spike-based computation. Extensive experiments demonstrate that the proposed models maintain strong performance across diverse NLP tasks while offering notable energy efficiency and neuromorphic suitability.

**Strengths:**

S1: This paper proposes a directly trained spiking language model that can be pre-trained with a decoder-only architecture, effectively shortening the spike length compared to ANN-to-SNN conversion methods.
S2: The paper completely removes the softmax operation and replaces it with discrete, binary-quantized computations.
S3: The overall design is relatively simple, and the writing is clear and easy to follow.

**Weaknesses:**

W1: The proposed softmax binarization function in this paper is similar to that of BiBERT (ICLR 2023). A more detailed comparison with related methods could be added.
W2: My main concern about this work is the experimental evaluation. In Table 1, the results of SpikeLM are directly cited; however, due to differences in training data and evaluation methods, such a comparison is not fair. Tables 2 and 3 mainly compare models trained on different datasets, making the conclusions less convincing. It is recommended to include more rigorous and comprehensive experiments to strengthen the claims.

**Questions:**

SpikingFormer is also a softmax-free method. When applying a causal mask to language data, what issues might arise? Would its performance be lower than the method proposed in this paper? (Experiments under the same conditions are needed to verify this.)

---

### Official Review · Reviewer_GRAm · 2025-11-05

**Soundness:** 2
**Presentation:** 2
**Contribution:** 2
**Rating:** 2
**Confidence:** 5

**Summary:**

This paper proposes replacing the energy-intensive softmax function in Spiking Transformers with a brain-inspired, softmax-free Winner-Take-All (WTA) mechanism. This allows for the creation of fully spike-driven language models, which are more energy-efficient and better suited for neuromorphic hardware. The authors introduce two novel, fully spike-driven architectures: (1)WE-SpikingFormer: An encoder model for masked language modeling, and (2) WD-SpikingFormer: A decoder model for causal language modeling. Experiments show that the WTA-based attention mechanism achieves performance nearly identical to that of standard softmax, validating it as an effective and energy-efficient substitute for language modeling.

**Strengths:**

1. Easy to follow
- The paper is well-written, and its arguments are clearly structured, making the proposed methodology and contributions easy to follow.

2. Fully Spike-Driven Architecture
- The whole architecture is spike-driven architecture by using membrane residual connections. This architecture is more efficient than spike-based residual connections.

3. Energy-Accuracy Trade-off
- The WD-SpikingFormer-0.4B model, for example, uses only 7% of the energy of the 1.5B-parameter DeepSeek-Distill-Qwen model while maintaining competitive accuracy on QAT benchmarks (28.4% vs. 33.2%).

**Weaknesses:**

1. Novelty
- The paper's claimed novelty rests on introducing the Winner-Take-All (WTA) mechanism, inspired by biological lateral inhibition . However, this WTA mechanism appears functionally similar to max pooling, which has already been utilized as an attention operation in prior works like MaxFormer [1] and SpikePool [2]. The paper does not clearly differentiate its approach from these existing max pooling methods. Furthermore, the paper does not seem to offer other significant novel contributions beyond the WTA mechanism.

2. Related works
- There is a significant overlap in content between the Related Work section and the "Introduction.

3. Unexplained Neuron Model Choice
- The paper applies the T-LIF neuron model to the WE-SpikingFormer and the NI-LIF(1x4) model to the WD-SpikingFormer. However, the text provides no justification for this specific design choice, leaving it unclear why two different neuron models were used for the two different architectures.

4. Scalability
- As the authors acknowledge in their limitations, the proposed models were only evaluated at the 0.4B parameter scale. It is therefore unclear whether the WTA mechanism, trained with a surrogate gradient, will remain stable and effective when scaled up to larger 7B+ parameter architectures.

[1] Fang, Yuetong, et al. "Spiking Transformers Need High Frequency Information." arXiv preprint arXiv:2505.18608 (2025).

[2] Lee, Donghyun, et al. "SpikePool: Event-driven Spiking Transformer with Pooling Attention." arXiv preprint arXiv:2510.12102 (2025).

**Questions:**

1. Could the authors elaborate on the core novelty of this paper? Additionally, the "Hard WTA" mechanism appears functionally equivalent to a global max pooling operation, which has been explored in other recent works.

2. The paper applies T-LIF neurons to WE-SpikingFormer and NI-LIF neurons to WD-SpikingFormer. Could the authors provide the rationale for this specific design choice? Furthermore, were any experiments conducted where these neuron models were exchanged (e.g., NI-LIF in WE-SpikingFormer and T-LIF in WD-SpikingFormer), and what was the resulting impact on performance?

3. The ablation study shows that Hard WTA ($k=1$), Top-k WTA, and Sparsemax yield almost identical results on CRT (43.2%, 42.9%, 43.2%, respectively), suggesting performance is insensitive to the exact sparsity. Did you investigate the effect of $k$ in Top-k WTA? Is there a performance change if $k$ becomes too large or too small?

---

### Official Review · Reviewer_HtL9 · 2025-11-07

**Soundness:** 3
**Presentation:** 2
**Contribution:** 2
**Rating:** 4
**Confidence:** 4

**Summary:**

To address the lack of softmax-based normalization in Spikeformer, the authors introduce a winner-take-all (WTA) mechanism into spiking Transformers and propose two softmax-free, spike-driven attention modules: WTA Spiking Self-Attention (WSSA) and Causal WTA Spiking Self-Attention (CWSSA).However, the current work has several shortcomings:

**Strengths:**

1. This direction is potentially meaningful and may alleviate Spikeformer’s limitations in fine-grained language modeling.
 2. It is also noteworthy that the experiments incorporate a decoder-style Spiking Transformer, which is relatively novel in this area.

**Weaknesses:**

1. The paper lacks a rigorous theoretical justification for why WTA can effectively replace softmax. Relying solely on biological inhibition as motivation is unconvincing, and no clear correspondence is established between WTA dynamics in neural circuits and attention weight allocation.
2. The design rationale of Hard WTA, Top-k WTA, and Sparsemax, as well as their compatibility with spike-based matrix operations, is insufficiently analyzed; these components are largely presented descriptively rather than being grounded in principled arguments.
3. Ablation studies confined to GLU are inadequate to support the claimed generality of the method; broader validation across diverse tasks and architectures is necessary.

**Questions:**

See weakness

---

### Author Response · Authors · 2025-12-02
**Official Comment**

We thank the reviewers for acknowledging the innovation and technical contributions of our proposed winner-take-all spiking transformer. We also sincerely appreciate their professional and insightful comments, highlighting the shortcomings in the related work review and the need for deeper methodological analysis.

We are grateful to all four reviewers for their constructive feedback, which will greatly facilitate the further improvement and refinement of our manuscript. Thank you once again!

---

### Note · Authors · 2026-01-15

**Comment:**

We would like to sincerely thank the four reviewers once again for their constructive comments.  We have further improved and refined the paper based on the reviewers’ feedback and are now withdrawing this submission to prepare the revised version for a future submission.

**Withdrawal Confirmation:**

I have read and agree with the venue's withdrawal policy on behalf of myself and my co-authors.